# Possible Gender Influence in the Mechanisms Underlying the Oxidative Stress, Inflammatory Response, and the Metabolic Alterations in Patients with Obesity and/or Type 2 Diabetes

**DOI:** 10.3390/antiox10111729

**Published:** 2021-10-29

**Authors:** Martha Lucinda Contreras-Zentella, Rolando Hernández-Muñoz

**Affiliations:** Departamento de Biología Celular y Desarrollo, Instituto de Fisiología Celular, Universidad Nacional Autónoma de México (UNAM), Apdo. Postal 70-243, Avenida Universidad # 3000, Coyoacán, Mexico City 04510, Mexico; mcontre@ifc.unam.mx

**Keywords:** type 2 diabetes, gender, oxidative stress, inflammation, metabolomics, obesity

## Abstract

The number of patients afflicted by type 2 diabetes and its morbidities has increased alarmingly, becoming the cause of many deaths. Normally, during nutrient intake, insulin secretion is increased and glucagon secretion is repressed, but when plasma glucose concentration increases, a state of prediabetes occurs. High concentration of plasma glucose breaks the redox balance, inducing an oxidative stress that promotes chronic inflammation, insulin resistance, and impaired insulin secretion. In the same context, obesity is one of the most crucial factors inducing insulin resistance, inflammation, and contributing to the onset of type 2 diabetes. Measurements of metabolites like glucose, fructose, amino acids, and lipids exhibit significant predictive associations with type 2 diabetes or a prediabetes state and lead to changes in plasma metabolites that could be selectively affected by gender and age. In terms of gender, women and men have biological dissimilarities that might have an important role for the development, diagnosis, therapy, and prevention of type 2 diabetes, obesity, and relevant hazards in both genders, for type 2 diabetes. Therefore, the present review attempts to analyze the influence of gender on the relationships among inflammatory events, oxidative stress, and metabolic alterations in patients undergoing obesity and/or type 2 diabetes.

## 1. Introduction

The most prominent causes of death in groups of patients with obesity are cardiovascular and cerebrovascular complications, such as infarction and stroke [1] because of an increase in adipose tissue, which is metabolically active, and the malfunction of leptin signaling which has been associated with the atherosclerotic process [2]. Insulin resistance is one of the major pathophysiological attributes of type 2 diabetes mellitus (type 2 diabetes) and is closely related to abdominal obesity and organ lipid deposition [3]. It is characterized by reduced sensitivity of insulinotropic tissues to the action of insulin, with a failure in triggering the insulin signaling cascade, which promotes glucose uptake in muscles, and adipose tissue and increases the activity of key glucose and lipid metabolism enzymes [4]. In obese individuals, expanded adipose tissue could present a lack of oxygen availability, which causes adipose tissue death, attracting macrophages that produce inflammatory cytokines, increasing insulin resistance [5].

An excess of liver triglycerides results in ineffective β-oxidation and the production of ceramides, diacylglycerols, and acylcarnitines. This accumulation of lipid derivatives leads to inflammation and contributes to liver insulin resistance. Increasing evidence has indicated that excessive triglycerides storage is a strong inducer of chronic inflammation [6]. Hence, it has been suggested that insulin resistance, as a major pathophysiological feature of both obesity and type 2 diabetes, could play a role in the pathology of Alzheimer’s disease, for its association with β-amyloid accumulation and hyperphosphorylation of the tau protein [7]. Thus, it is very likely that some pathological processes in the periphery, including inflammation, are associated with brain pathology, and this has led to the hypothesis that treatments for obesity and type 2 diabetes readily reduce the development of neuro-inflammation, probably diminishing the onset of neurodegenerative diseases (Figure 1). Among the reported treatments are, in addition to anti-diabetic drugs, the incretin hormone therapy [8] or application of the fibroblast growth factor-21 [9].

An Increase in fat intake leads to adipocytes becoming hypertrophic and to an accumulation of natural killer (NK) cells. The hypertrophic adipocytes and NK cells produce pro-inflammatory cytokines, which provide a chemotactic gradient for recruitment of other adaptive and innate immune cells. These pro-inflammatory molecules activate stress kinases such as I*κ*B kinase (IKK) and Jun kinase (JNK), leading to insulin resistance. The inflammatory process and insulin resistance induce metabolic consequences in the liver (increased Kupffer cells), pancreas (accumulation of macrophages), in the brain the microglia become more inflammatory (accumulated macrophages), and gut (changes in immune populations and microflora).

The metabolic syndrome is recognized as a conjunct of risk factors that can predispose an individual to cardiovascular diseases and type 2 diabetes, among other pathologies. Impaired insulin-mediated glucose uptake is the principal abnormality that bridges the metabolic and hemodynamic disturbances found in the metabolic syndrome [10]. Impaired beta-cell function along with insulin resistance can appear years before the development of diabetes and is already present in most pre-diabetic patients [11]. Insulin resistance can lead to increased systemic blood pressure, elevated triglyceride levels, and lowered HDL levels, which can induce a macrovascular dysfunction; here, oxidative stress and inflammation promote micro-angiopathic changes that are already present in pre-diabetic states [12].

Hyperglycemia induces oxidative stress and upregulation of pro-inflammatory factors, promoting a vascular dysfunction (Figure 2). Therefore, oxidative stress may be involved in the progression from prediabetes to diabetes, impairing glucose uptake in muscle and fat cells and decreasing insulin secretion from β-cells [13]. Reduction of systemic oxidative stress through the use of NADPH-related metabolic inhibitors readily improves glucose metabolism in a mouse model [14].

Binding of insulin stimulates the association of its receptor with downstream mediators, including insulin receptor substrate-1 (IRS-1) and PI3K. The PI3K/AKT/GSK-3β signaling pathway is involved in insulin signaling transduction, and GSK3β is regulated and controlled by insulin in this signaling pathway, which is related to glycogen synthesis regulation. Through specific receptors, hyperglycemia, fatty acids, and pro-inflammatory cytokines, as well as ROS, activate JNK and IKKB favoring NF-κB translocation into the nucleus and, therefore, transcription of inflammatory genes.

Metabolomics, as the systematic analysis and study of metabolites in a biological sample, including low-molecular-weight biochemical compounds such as organic acids, amino acids, sugars, lipids, and nucleotides [15], has been utilized to assess correlations among metabolites, risk for type 2 diabetes and its symptoms [16]. Metabolomics and metabolic features interaction studies have expanded the current literature, strengthening the knowledge of the pathophysiological pathways underlying type 2 diabetes, and have allowed the identification of novel and reliable biomarkers that can markedly improve the diagnosis and treatment of type 2 diabetes (Figure 3). Therefore, the present review was aimed to correlate type 2 diabetes with inflammatory events, oxidative stress, and changes in the metabolic pattern that participate in the physio-pathology of the disease. In particular, we tried to analyze the influence of gender in these biomedical aspects. For this, we will discuss firstly the relations of obesity with inflammatory states and with insulin resistance in Section 2 and Section 3, to focus on oxidative stress, type 2 diabetes, and a putative influence of gender (Section 4) to finally comment on interactions of inflammation and metabolic regulation in type 2 diabetes (Section 5).

Reactive oxygen species (ROS) play a central role in the interactions involving inflammation, oxidative stress, and metabolic control. Hyperglycemia increases ROS production and chronic inflammation. Altered expression of miRNAs and epigenetic regulation of oxidative stress genes are contributing factors to hyperglycemic memory. Sources of ROS include NADPH oxidase, dysfunctional endothelial nitric oxide synthase (eNOS), and xanthene oxidase. Excessive production of ROS can exacerbate and contribute to the pathogenesis of insulin-resistance and impaired insulin secretion. In addition, high levels of branched-chain amino acids (BCAAs) are often associated with increased BMI levels and insulin resistance whereas low levels of BCAAs are accompanied by lower BMIs and good insulin sensitivity.

## 2. Obesity and Associated Pathologies Linked to Inflammation

Different pathologies, such as cardiovascular diseases, neuro-inflammation, resulting in neurodegenerative diseases, metabolic syndrome, and type 2 diabetes, have been associated with increased secretion of inflammation-promoting cytokines. Inflammatory cytokines are produced by different cell types and secreted into the circulation, where they regulate different tissues through their activities. There has been a great interest in determining the metabolic consequences of obesity derived from the current diet, in which foods are usually rich in carbohydrates and fats. Among the most important results obtained was the relationship observed between obesity and inflammation. The specific factors that initiate inflammation are not known completely yet and, probably, differ among diverse organs and tissues, but are associated with the activation of the innate immune system in response to cellular homeostasis alterations [17,18,19].

Obesity produces a marginal increase in the levels of the C-reactive protein and inflammatory factors that disturbs the insulin sensitivity in diverse organs, such as the pancreas, liver, heart, brain, and tissues like the adipose tissue and skeletal muscle. This type of inflammation is further linked to the development of insulin resistance [20,21] and can lead to fibrosis or necrosis [22]. Obesity and insulin resistance are major risk factors for type 2 diabetes (Figure 1).

Evidence has revealed that excessive triglycerides storage is a strong inducer of chronic inflammation [6,7,23], and chronic inflammation may be one of the possible mutual denominators of metabolic pathologies. In obese people, for example, triglycerides, normally present in adipose tissue, are accumulated in the liver producing non-alcohol fatty liver disease (NAFLD). This pathology causes a non-productive β-oxidation and accumulation of ceramides, diacylglycerols, and acylcarnitines, producing inflammation and resulting in liver insulin resistance [24]; progress of NAFLD can develop to nonalcoholic steatohepatitis and, occasionally, to cirrhosis, as a result of inflammation and fibrosis [25] (Figure 1).

Besides, obesity provokes metabolic stress. The Kupffer cells, which are liver-specific macrophages, are stimulated, and leukocytes are incorporated to the damaged liver. These situations promote the secretion of chemokines and cytokines, mainly TNF-α and IL-6 [26]. The resulting inflammation induces dilatation of adipocytes and mechanical stress [27]. In the same way, studies in obese subjects have reported a direct relationship between inflammation and insulin resistance. Very important signaling routes are involved, like those in which NF-κB and JNK participate [28], in addition to the proinflammatory molecules, scaffolding proteins, and cytokines like TNF-α. The pentatricopeptide repeat (PRR) proteins, which belong to the family of leucine-rich repeat-containing (NLR) proteins and detect endogenous ligands produced by obesity, stimulate the cryopyrin/NLR pyrin domain-containing 3 (NLRP3) inflammasome, inducing the synthesis of IL-1β and IL-18 through caspase-1 in macrophages [29]. PRRs’ stimulation could regulate the production of lipids such as ceramides and sphingolipids, which stimulate inflammation, inhibiting the capability of saturated fatty acids to induce insulin resistance [22,30,31,32]; many of these pathways can interfere by blocking insulin action (Figure 1). The arachidonic acid–derived product, leukotriene B4, is released from adipocytes and recruits macrophages to the adipose tissue, attenuating insulin signaling in myocytes and hepatocytes [33]. Similarly, galectin-3, which belongs to the lectin family, is produced by macrophages; this protein is associated with inflammation and could provoke insulin resistance in muscle, liver, and fat cells, blocking insulin receptor signaling [34]. Galectin-3 is involved in broad biological functionality [35] and has been demonstrated to be involved in cancer, inflammation, and fibrosis [36].

On the other hand, the adipose tissue does not only function as energy reserve but also has an endocrine function because it is formed of various types of cells (adipocytes, pre-adipocytes, fibroblasts, and immune-competent cells) [1,37,38]. Recently, it has been hypothesized that the adipose tissue is a key tissue in the induction of inflammation, due to information sharing between the different types of cells that compose it and other immune cells that infiltrate it. Adipokines are adipocyte-specific cytokines, responsible for the energetic balance in the body. Among adipokines, the anorexigenic leptin secretion has pro-inflammatory consequences, whereas adiponectin has an anti-inflammatory effect and raises insulin sensitivity [39]. In obese subjects, the anorexigenic leptin is increased, which can be associated to insulin resistance [40,41].

Furthermore, in obese subjects, the accumulation of triglycerides increases adipose tissue mass and induces hypoxia, which can lead to cellular death; the adipocyte remnants attract macrophages. Macrophages are cells with diverse activities; in addition to having chemotactic activity, they act as antigens, secrete cytokines that are involved in cellular metabolism, very importantly, in lipid metabolism. The release of cytokines leads to inflammation, which has been postulated to be associated with predisposition to various pathologies. Macrophages undergo drastic alterations when obesity occurs. The amount of macrophages increases due a greater number of M1-polarized macrophages, which are pro-inflammatory and produce cytokines, such as TNF-α, and start the NF-κB and JNK pathways. Consequently, the ratio of M1 to M2 macrophages increases, which is considered a characteristic of the inflammation occurring in obesity and adiposis, as well as in insulin resistance and metabolic disease [42]. The immune response in obesity starts in the adipose tissue, participating T effector cells, B cells, NK cells, and others that secrete cytokines and promote inflammation [18]. Similarly, the elevated levels of free fatty acids, insulin, and glucose, observed in insulin resistance, induce macrophages activation. The macrophages remain permanently activated and promote chronic inflammation and secretion of the anorexigenic leptin [43,44].

Free fatty acids can also promote inflammation by collateral incorporation to TLR4 and TLR2 through the adaptor protein fetuin A, activating NF-κB and JNK1 [32]. In the same way, pro-inflammatory cytokines participate in insulin resistance in the adipose tissue via the activation of IκB kinase and c-Jun N-terminal kinase, responsible for dephosphorylating the serine of the insulin receptor substrate-1 (IRS-1), turning insulin into less efficient [45]. These results confirm that obesity and metabolic syndrome co-occur with chronic subclinical inflammation that begins early in the adipose tissue [22] and then affects all cell homeostasis (Figure 1).

Research about obesity has evidenced that intestinal absorption is augmented, increasing circulation of Lipopolysaccharide (LPS) from the intestinal Gram-positive bacteria [46]. Given that LPS is a systemic circulating factor, it has been postulated that it might function as an amplifier of the inflammatory pathway, rather than as tissue-specific that starts the inflammation mechanism. Inflammation starts with the activation of receptors like TLR4 in fat cells and interrelated with type 2 diabetes in humans [43].

## 3. Obesity and Insulin Resistance

The signaling cascade of insulin is not fully understood yet. It is an intricate signaling pathway, with numerous regulatory proteins participating in it. If some of these proteins fail in their function, some tissues may become insulin resistant [47], which is associated with metabolic pathologies like obesity. In obese subjects, numerous inflammatory signals act together to stimulate serine kinases that interfere with insulin signaling. Several kinases, including, most probably, different kinases activated by stress and which can be dependent and independent of insulin, can block the insulin receptor signaling by promoting the phosphorylation of serine/threonine of insulin receptor substrates and reducing tyrosine phosphorylation; all these modifications affect downstream insulin signaling [48]. Normally, insulin binds to its tyrosine kinase receptor; tyrosines are auto-phosphorylated and initiate the signaling cascade (Figure 2). The signaling pathway includes participation of Shc (adaptor protein family) and APS proteins (adaptor molecule containing PH and SH2 domains) [49], which favor the activation by the phosphorylation of IRS-1 through the activity of kinases.

Among the kinases induced by insulin are protein kinase C, serine/threonine-protein kinase 2 (AKT2 or Akt-2; activated by phosphatidylinositol 3,4,5-trisphosphate), S6K1 (which is implicated in mTOR pathway activation), extracellular signal-regulated kinase 1/2, and ROCK1 [49]. Phosphorylation of Akt-2 regulates various proteins downstream, such as rapamycin (mTOR) and glycogen synthase kinase-3β [50,51]. Other kinases independent of insulin, like AMPK and GSK3, phosphorylate IRSs, and the pathway is activated downstream [52].

Therefore, the activated IRS-1 generates signaling by attaching and stimulating phosphoinositide-3 kinase (PI3K), the site in the pathway at which insulin and leptin intersect. In the case of obese subjects, chemokines participate in liver insulin resistance by activating the IκB kinase [53]. This enzymatic complex is upstream in the NF-κB signal transduction pathway and participates in the cellular reaction to inflammation. The IκB kinase activation depresses IRS activity and, if leptin resistance is present, it can promote insulin resistance by depressing PI3K activation [54]. Furthermore, if IRSs are phosphorylated at particular sites, like serine 307, by some kinases such as IKKβ/NF-κB and c-Jun N-terminal kinase (JNK), their signaling activity in this via is reduced and insulin resistance occurs [55]. On the other hand, tyrosine phosphatase 1B (PTP1B) functions as a negative regulator of the signaling cascade of insulin, by dephosphorylating tyrosine residues of IRS-1 [56] (Figure 2).

As described above, inflammation and the consequent presence of cytokines, like the tumor necrosis factor-α, monocyte chemotactic protein-1, C-reactive protein, and interleukins, are caused by obesity and are strongly associated with insulin resistance [44,57]. For example, TNF-α interferes with insulin signaling through serine phosphorylation of IRS-1, which decreases GLUT-4 expression and, consequently, diminishes glucose entry into cells [58]. In addition, inflammation releases nitric oxide, which inhibits the PI3K–Akt pathway through S-nitrosylation of Akt [59]. Besides, as previously discussed, obesity is associated with the presence of inflammation and insulin resistance and these alterations are, in turn, associated with oxidative and nitrosylative stresses [13,60,61] (Figure 2).

## 4. Oxidative Stress, Insulin Resistance, Type 2 Diabetes, and Gender

### 4.1. Obesity and Oxidative Stress

In normal conditions, cell metabolism produces free radicals, which exert a crucial function in signaling; a disequilibrium in free radicals (augmented free radicals in addition to decreased antioxidants) induces oxidative stress (reactive oxygen species, ROS) [61]. Food with elevated calories and rich in fats cause obesity, metabolic syndrome, insulin resistance, and type 2 diabetes; consumption of these types of food increases the presence of p47phox, IL-6, and ROS production [62].

As previously mentioned, obesity produces inflammation, which in turn is associated with insulin resistance, leading to metabolic syndrome, finally, inducing type 2 diabetes. Type 2 diabetes is currently a pathologic condition widely distributed worldwide, and it has been suggested that its associated damage may be different in relation to age and gender of the diabetic subjects. The cultural differences, lifestyle, behavior, environment, nutrition, genetics, epigenetics, metabolic and hormonal level dissimilarity between men and women are the reasons for the diversity in predisposition and development of type 2 diabetes between them. The symptoms, risks, and complications observed in them are also different.

Oxidative stress induces insulin resistance by disturbing the insulin signaling pathway, affecting regulation at the level of adipokines [63] and favoring some of the mentioned serine–threonine kinases pathways. Expressions of factor NF-kβ [64], TNF-α, IL-1β (which participate in the β cells injury), plasma endotoxin, and Toll-like receptor (TLR)-4 are increased. Furthermore, suppression of cytokine signaling 3 and IKKβ in association with TNF-α, NF-κB, and JNK affect the IRS-1 (ubiquitylation and proteolysis) by inhibiting its phosphorylation [65] (Figure 2).

Regarding glucose tolerance, altered fasting glucose levels are predominant in men and altered glucose tolerance is common in women. Usually, normal glucose tolerance is preserved if insulin secretion is capable to offset the diminution of insulin sensitivity, which is present in increased body mass index (BMI) or age [66]. When type 2 diabetes is present, the deficiency in insulin sensitivity and insulin secretion is significant and comparable in both genders; insulin sensitivity is reduced when BMI increases in an equal proportion for both men and women. Indeed, during aging, a deficiency in glucose tolerance is observed and the differences between genders decrease, although the metabolic profile is affected more in women than in men [67,68], promoted by insulin resistance and associated with the weight gain observed in type 2 diabetes [69,70].

In association with insulin resistance, in obese subjects, the presence of biomarkers of oxidative stress has been observed; the latter include malondialdehyde (indicative of lipid peroxidation and ROS), protein carbonyls, 4-hydroxy-2-nonenal, hydroperoxides, protein oxidation products, 3-nitrotyrosine, advanced glycation end products (AGEs), carbohydrate metabolites, and 8-hydroxy-2′-deoxyguanosine (8-OH-dG), which have the ability to diminish insulin sensitivity [13,71,72] (Figure 2). Obesity is associated with a higher production of H_2_O_2_ by mitochondria and a decrease in the reduced glutathione/oxidized glutathione disulfide in skeletal muscle [73], which has a negative correlation with body mass index. A rise in the oxidative redox state produces the increase in the levels of the transcriptional factor mentioned above [74].

Many women present being overweight or obese after 45 years of age, in association with pre-menopause and menopause. These stages in the life of women are associated with hormonal changes (decrease in the concentration of estrogens, for example); the decrease in estrogens favors weight gain and type 2 diabetes in women. In men, overweight is observed at younger ages [75]; in both, obesity is the main risk factor for type 2 diabetes [76]. Moreover, disturbances in glucose tolerance are more frequent in women than in men no matter the age [76]. The risks associated with type 2 diabetes seem to be lesser in women than in men; this difference generates a greater gap regarding central adiposity, coagulation, and inflammation alterations between diabetic and non-diabetic women than in men [69]. Regarding adipokines, women display increased expression of leptin and adiponectin and their receptor in abdominal adipose tissue. Overall, meta-analyses have revealed that women present higher leptin and adiponectin levels than men, with similar age and BMI, which could be associated with their sex hormones [77]. In some longitudinal studies, augmented plasma leptin is associated with a higher diabetes risk in men [78]. Furthermore, an inverse correlation between plasma adiponectin levels and insulin sensitivity in obese and diabetic subjects has been observed, which has been postulated to be higher in women [79,80,81]; besides, androgens could provoke a diminution in adiponectin secretion [78].

### 4.2. Type 2 Diabetes and Gender

Gender and biological sex influence several pathologies, among which are metabolic disarrays, like insulin resistance and diabetes. In general, diabetes is predominant in men in relation to women, especially in subjects between 35 and 69 years of age, who are considered to be middle-aged; the top in diabetes incidence happens earlier in men (65–69 years) than in women (70–79 years) [82]. In association, in almost all animal models, males are more likely to present obesity, insulin resistance, and hyperglycemia than females due to nutritional defiance [83]. Insulin resistance is observed when peripheral tissues (adipose, skeletal muscle, and liver) do not react correctly to insulin, affecting the uptake of glucose. The organs and tissues that participate in glucose metabolism express and respond to inflammatory mediators; insulin resistance, in turn, is associated with a slight inflammation state [84]. Estrogens are involved in the regulation of metabolic processes and can affect inflammatory responses. Macrophages and monocytes associated with inflammation are activated by estrogen through their receptors. Additionally, there is a relation between the decreased levels of estrogen in post-menopausal women and an augmented inflammatory state. Post-menopausal women have augmented lymphocyte and monocyte counts and enhanced expression of pro-inflammatory cytokines. These results suggest that estrogens influence insulin resistance [85]; in addition, insulin resistance can promote the appearance of metabolic syndrome and, eventually, of type 2 diabetes. Men are more vulnerable to metabolic syndrome than premenopausal women; the vulnerability is modified when estrogen levels are reduced during menopause, regardless of age [86].

It is currently clear that many aspects of energy balance and glucose metabolism are regulated differently in men and women and influence their predisposition to type 2 diabetes. When women are reproductively active, they have energy requirements partitioning very different from those of men, such as carbohydrate and lipid utilization as energy sources, which favor energy storage in subcutaneous adipose tissues and preserve them from visceral and ectopic fat accumulation. Insulin sensitivity is greater in women, who are characterized by higher capacities for insulin secretion and incretin responses than men; however, if glucose tolerance is present, both gender progress toward diabetes. On the other hand, there are many suggestions regarding the protective actions of endogenous estrogens, principally through the activation of the estrogen receptor α in several organs, like the brain, liver, skeletal muscle, adipose and pancreatic tissues [83]. Clinical and experimental studies show that post-pubertal sex steroid hormones essentially provide for sex differences in diabetes vulnerability. The protection offered by endogenous estrogens to women is confirmed by the loss of this protection in relation to glucose homeostasis during menopause. The decrease in estrogens in menopausal women results in an increased occurrence of metabolic disorders, such as type 2 diabetes. Ageing prompts body changes in both sexes; menopause prompts the progressive increase of visceral fat, which contributes to the augmented possibility of metabolic disorders [87,88]. Therefore, it is very important to bear in mind the sex differences, in both human and animal models, in preclinical and clinical investigations [89].

On the other hand, among the organs involved, the liver exerts an important role in glucose homeostasis, since gluconeogenesis and glycogenolysis take place in this organ. Regarding liver metabolism, augmented liver enzymes (alanine aminotransferase, aspartate aminotransferase, and γ-glutamyl transferase) have been observed before the occurrence of type 2 diabetes in both genders. A strong relationship between type 2 diabetes and this enzyme has been observed, which can be explained by the relation among γ-glutamyl transferase, fatty liver, oxidative stress, and, hence, with insulin resistance compared with the other enzymes [89]. According to [90], the fatty liver index, liver enzymes, triglycerides, waist circumference, and BMI are better indicators of metabolic syndrome in women than in men.

In the same context, there is evidence of a putative role of erythrocytes in a differential management of nitrogen-related metabolites, which occurs in healthy women and men [91]. An increased activity and expression of arginase I seem to be associated with a diabetes-induced increase in oxidative stress, which also initiates the feed-forward cycle of diminished nitric oxide (NO) levels and oxidative stress [92]. Moreover, citrulline could promote NO production and endothelial function and enhance peripheral insulin sensitivity [93], improving organ perfusion and endothelial metabolism, which might involve an antioxidant property [94]. Indeed, it has been reported that there are differences in nitrogen metabolism in humans according to gender. Changes in vascular NO activity may contribute to changes in cardiovascular risk associated mainly with men, probably related to the α-adrenoreceptor responsiveness, among other mechanisms [95]. In addition, increased ammonia production consequent to metabolic pathologies has been found mainly among middle-aged Japanese men with chronic liver disease rather than in women suffering the same hepatic disturbances [96].

A recent study from our group evidences that in healthy subjects there are significant gender differences in NO production and in arginine metabolism [91]. For instance, the arginine/nitrites and arginine/citrulline ratios are bigger in erythrocytes of healthy men than in those of women, suggesting a lower flow of arginine through putative nitric oxide synthase in men; healthy men also seemed to use less arginine as substrate for arginase, i.e., the arginine/urea ratio [91]. These data are consistent with the previously mentioned suggestion that women appear to have a more active NO metabolism than men. It has also been observed that the onset of type 2 diabetes is capable of canceling out gender differences in managing nitrogen-related metabolites [91]. Therefore, a possible role of erythrocytes as an extrahepatic mechanism controlling serum levels of nitrogen-related metabolites can be suggested, which differs according to sex in healthy subjects, whereas the serum control is lost in diabetic subjects. All possible differences attributable to gender in the development of type 2 diabetes are shown in Table 1.

## 5. Inflammation and Metabolic Regulation in Type 2 Diabetes

The metabolomics approach constitutes a systematic analysis and study of metabolites in a biological sample [14], including biochemical compounds, such as organic acids, amino acids, sugars, lipids, and nucleotides [97]. It allows evaluating feasible associations between specific metabolites, such as amino acids and type 2 diabetes, with subsequent analyses of human studies on the utility of some metabolites, including glutamine, glycine, and aromatic amino acids, as reliable biomarkers [98] (Figure 3).

In addition, the microbiota’s composition can play a major role in the development of obesity and diabetes, and some treatments directed against diabetes may have actions mediated by gut bacteria (Figure 1). Several underlying mechanisms could include an association of reduced and altered microbial diversity with inflammation, insulin resistance, and adiposity [98]. In particular, changes in the diversity and enrichment of different bacterial phyla are related to the inflammation grade and the ability of harvesting energy from food, where a high-fat diet favors the growth of bacteria capable of beneficial actions on energy extraction (Figure 1). Indeed, a massive oversupply of glucose into the pentose phosphate pathway of intestinal bacteria can be assumed to result in elevated levels of erythrose 4-phosphate [98] (Figure 3). In fact, dyslipidemia is an independent risk factor for type 2 diabetes [99,100], which includes total lipid or lipid class (i.e., triacylglycerols or HDL) levels. Recent studies have identified signatures of particular lipids or patterns in lipid classes to be predictive of diabetes onset. In the “Framingham Heart Study” cohort, it was conclusively identified that saturated or monounsaturated fatty acids of lower carbon number are associated with an increased risk of type 2 diabetes [101] (Figure 3).

Fatty acids can exert inflammatory effects on macrophages, which could contribute to inflammation [102]. After entering the cell, fatty acids are thio-esterified into their acyl-CoA derivatives, catalyzed by long-chain acyl-CoA synthetases (ACSLs). These observations indicate that ACSL1-derived lipids, not glucose, play a critical role by promoting the inflammatory phenotype of macrophages associated with diabetes. Additionally, recent exciting discoveries link intestinal microbiota metabolism of dietary-derived saturated fats to cardiovascular disease risk, highlighting these parameters as attractive potential therapeutic targets for obesity/diabetes [103,104] (Figure 3).

### Metabolites, Metabolomics, and the Pathogenesis of Type 2 Diabetes

Type 2 diabetes can be considered a nutritional disorder characterized by the inability of the body to respond to insulin, which leads to many complications, including kidney failure, retinopathy, lower-limb amputation, an increased risk of cardiovascular disease [105], and stroke [106]. Several risk factors have been associated with type 2 diabetes, including obesity, high cholesterol and blood sugar levels, family history of type 2 diabetes, and history of gestational diabetes [107,108]. In addition, the concentrations of branched-chain amino acids (BCAAs) have been recorded higher in patients with diabetes than in healthy subjects, results that are reproducible and statistically significant [14,97,109]. Moreover, the relationship between BCAAs and type 2 diabetes has been explored, focusing on the total concentration of all three BCAAs, i.e., leucine, isoleucine, and valine (Figure 3). Another study correlated high concentrations of BCAAs with insulin resistance, presented by the experimental subjects receiving an amino acid-enrichment diet [109]. Based on the results, the authors concluded that an increased concentration of BCAAs is a reliable predictor of future insulin resistance among patients with type 2 diabetes, proposing these associations as targets for clinical management of the disease [110,111,112,113]. For instance, a study reported higher plasma levels of branched-chain and aromatic amino acids and higher glutamate-to-glutamine ratios in patients with diabetes than in healthy individuals [109]. These findings agree with studies reporting a positive association between obese individuals and elevated BCAAs and glutamate levels (Figure 3).

Other amino acids, such as hydroxyl acids and hydroxybutyrate, seem to induce an elevated insulin resistance and impaired glucose tolerance in patients with type 2 diabetes, and β-hydroxybutyrate and 3-hydroxybutyrate could correlate with a higher risk of prediabetes [114,115,116,117]. With regards to changes in lipids metabolism, some studies have focused on relationships between the metabolomic lipid profiles and type 2 diabetes, using high-throughput techniques to identify various classes of lipids, such as plasma phospholipids, triglycerides, sphingolipids, and glycerophospholipids [118]. Through these methods, high concentrations of glycerophospholipids and sphingomyelins have been found in individuals with type 2 diabetes. In addition, there are reports indicating that increased levels of fatty acids, such as dodecanoic and myristic acids, occur in patients with diabetes [119]. Therefore, efforts have been made to identify these lipid profiles that may facilitate prediction and management of the disease [14,119]. In addition, organic acids like acetic acid, dimethyl ester, and maleic acid have been associated with type 2 diabetes; other organic compounds, such as purines and arginine, citrulline, and ornithine, are also altered at the onset of diabetes [91,113,120].

Therefore, the combination of glucose, leucine, and other activators stimulates the mammalian target of the rapamycin complex (mTORC) pathway, inducing the proliferation of β-cells and insulin secretion. Leucine can increase the activity of the mTOR pathway, resulting in activation of S6 kinase and leading to inhibition of IRSs through serine phosphorylation. Augmented activity of the mTOR complex could promote an inability of β-cells to release insulin through an inhibitory effect on the mentioned kinase, leading to cellular insulin resistance and contributing to the development of type 2 diabetes (Figure 2). For instance, the BMIs of subjects have been compared to the concentrations of BCAAs in their diets and their insulin resistance levels. Obese subjects had higher metabolic concentrations (signatures) of BCAAs and higher resistance to insulin than lean individuals with a lower BMI. In addition, rats fed a diet rich in BCAAs display increased rates of insulin resistance and impaired phosphorylation of enzymes within their muscles. In contrast, obese mice with pre-diabetes fed a low-BCAAs, calorie-unrestricted, high-fat, and high sugar diet experience an improvement in metabolic health [121] (Figure 3).

The pathway involving BCAAs (essential amino acids for humans) is similar and begins with pyruvic acid, its production is increased when high glucose levels are available. Furthermore, in the presence of high glucose levels, more pyruvic acid is available to enteric bacteria, and more pyruvic acid is subsequently produced. Together with pyruvic acid, high amounts of the resulting amino acids, valine, leucine, and isoleucine (BCAAs), are produced. Metabolites such as glucose, fructose, amino acids, and lipids, which are typically altered in individuals with type 2 diabetes, can be identified by metabolomic techniques and used as potential diabetes biomarkers.

Other groups have also identified BCAAs and aromatic amino acids as predictors of type 2 diabetes in both human and animal models [122,123,124]. Further work with the “Framingham cohort” identified 2-aminoadipic acid (2-AAA) as an independent biomarker for risk development and highlighted the role of 2-AAA as an insulin secretagogue [125]. The 2-AAA is an intermediary metabolite of lysine degradation and has previously been shown to be increased by diabetes and renal failure [126] and has been suggested to be a biomarker of oxidative stress [127,128].

Identification of these biomarkers provides insight into the pathogenesis of diabetes. For instance, the increase in BCAAs may impact insulin sensitivity through the mTORC, as BCAAs could activate mTORC1 and the downstream target ribosomal protein S6 kinase 1 (S6K1) [129]. Additionally, catabolism of BCAAs can provide intermediates for the TCA cycle, potentially driving energy production [130]. The idea that the TCA cycle flux is altered in diabetes has been supported in other metabolomic studies in rats and mice [131,132]. Besides, reduced blood concentrations of glycine can also function as a predictor of type 2 diabetes [133] (Figure 3).

Other studies have been addressed at determining blood metabolite profiles before and after glucose loading [133,134], differentiating responses in obese from those of lean individuals [135]. These studies have noted differences in levels of acylcarnitines, glutamine/glutamate, additional amino acids, and other small molecules. These observations raise the possibility that alterations in plasma metabolite levels could be good predictors of the onset of diabetes and, therefore, aid in the identification of ‘at risk’ individuals by adding information over standard clinical markers. In this context, a study reported that serum indoxyl sulfate correlates inversely with renal function and might have a direct relationship with aortic calcification [136].

## 6. Conclusions

It is likely that the initial trigger of metabolic inflammation is the disruption of energy homeostasis produced by a positive energy balance and, therefore, the initial response tries to relieve the anabolic pressure produced by obesity. In time, this initial adaptive response fails and is transformed to a maladaptive one that can perpetuate inflammation in obese and type 2 diabetes patients. Although obesity and type 2 diabetes have a multifactorial origin, it is obvious that insulin and leptin resistance are deeply involved in these pathologies. In the brain, activated microglia produce pro-inflammatory cytokines in response to various stimuli, such as oxidative stress. The resulting neuro-inflammation and its molecular and cellular mechanisms are most likely related to obesity and type 2 diabetes. Early diagnosis and treatment of hyperglycemia should be clearly performed years before the development of type 2 diabetes (pre-diabetic patients), and oxidative stress and inflammation during prediabetes could be useful markers for clinicians to prevent its progression to type 2 diabetes. Changes in plasma metabolites, identified through metabolomic techniques, can also be useful as type 2 diabetes biomarkers. Targeted immunotherapies and anti-hyperglycemic medication are gaining a relevant role in improving the inflammatory state in patients with diabetes.

However, there is an evident lack of information about the influence of gender on the physiopathology of type 2 diabetes, as well as on its progression. Clinical parameters, such as fatty liver index, liver enzymes, triglycerides, waist circumference, and body mass index, are better indicators of metabolic syndrome in women than in men, and we have found significant differences in erythrocyte’s metabolism of nitrogen-related compounds have been identified in women and men with type 2 diabetes. Nonetheless, much research is still needed to evaluate this putative influence of gender in the future therapeutics for T2DM.

## Figures and Tables

**Figure 1 antioxidants-10-01729-f001:**
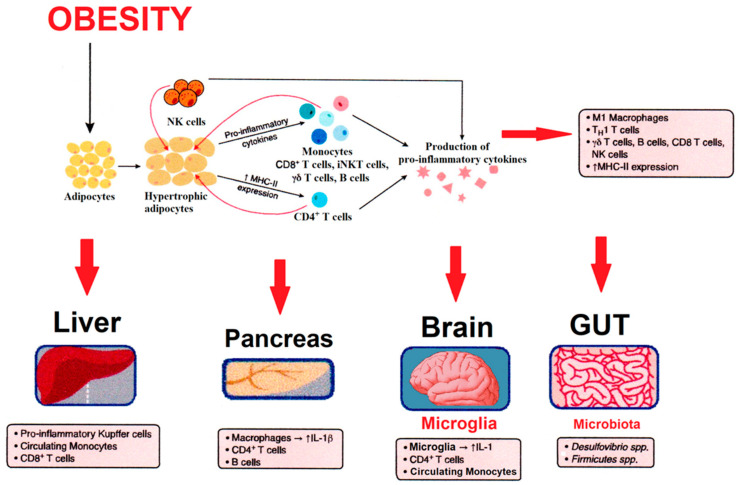
Scheme of peripheral inflammation connectivity involving obesity-Figure 1 induced insulin resistance in the adipose tissue in obesity and type 2 diabetes mellitus.

**Figure 2 antioxidants-10-01729-f002:**
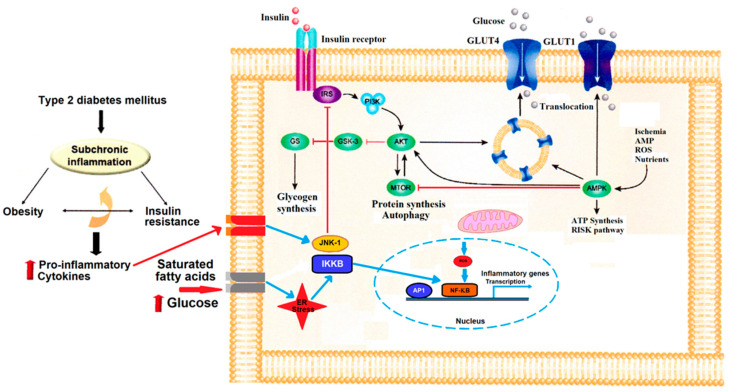
The insulin signaling pathway and the molecular mechanism of insulin resistance due to inflammation.

**Figure 3 antioxidants-10-01729-f003:**
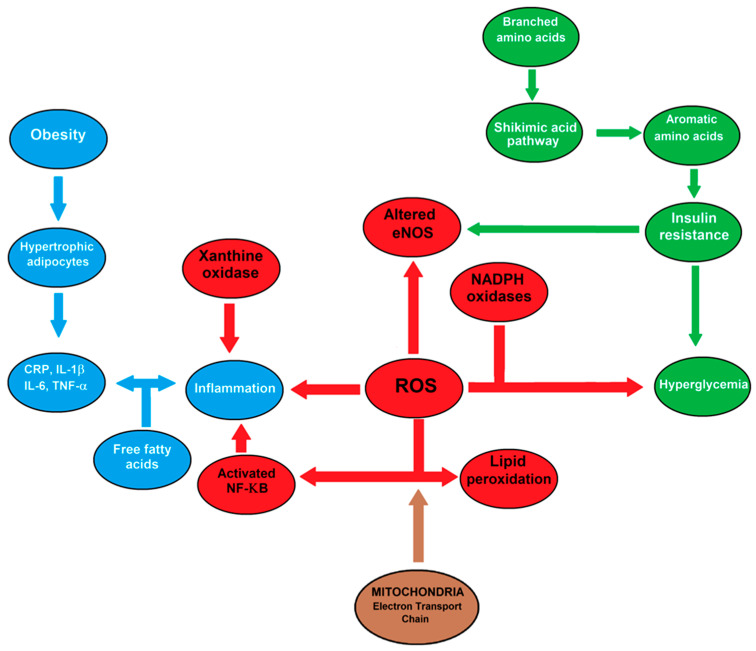
Major interactions among oxidative stress, inflammation, hyperglycemia, insulin-resistance, and metabolic changes in patients with type 2 diabetes.

**Table 1 antioxidants-10-01729-t001:** Some gender differences in metabolic regulation, parameters of inflammation and oxidative stress, as well as diabetes susceptibility.

Parameter	Finding	Sex Difference	Reference
Insulin resistance	Onset of T2DM	Predominant in men (35–69 years old)	[82]
Insulin resistance	Inflammatory state	Women show a larger state	[84]
Insulin resistance	Metabolic syndrome	Men show a higher vulnerability	[86]
Insulin sensitivity	Insulin secretion	Higher in women	[83]
Liver metabolism	Increased enzyme activities	Similar in both sexes	[89]
Metabolic syndrome	Indicators of liver function	Better indicators of metabolic alterations in women	[90]
Nitrogen metabolism by erythrocytes	Arginine catabolism and ammonia	More increased in diabetic men	[91]
Cardiovascular risk	Vascular NO activity	More diminished in diabetic men	[95]
Chronic liver disease	Pathological ammonia production	More frequent in middle-age men	[97]
Blood arginine metabolism	Increased arginine flow in healthy men	Diabetes abrogates gender differences	[91]

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
