# Peer review of "Possible Gender Influence in the Mechanisms Underlying the Oxidative Stress, Inflammatory Response, and the Metabolic Alterations in Patients with Obesity and/or Type 2 Diabetes"

_antioxidants, 2021, doi:10.3390/antiox10111729_

Round 1

Reviewer 1 Report

N/A

Author Response

Thank you

Reviewer 2 Report

The text of the review has been carefully revised and implemented according to the suggestions of the Referee, and is now suitable for publication in "Antioxidants"

Author Response

Thank you

Reviewer 3 Report

The authors tried to improve the manuscript.

Here are my comments,

Major:

  1. The manuscript is not focusing on the gender differences but on inflammatory mechanisms in obese and T2DM. So the title needs to be changed upon its consequence. Abstract and conclusion also need to be changed.
  2. In many paragraphs, the authors keep repeating the same phrase. It is the phrase saying inflammatory response-related signaling cascade. Repetition the fact we already know makes the readers being out of focusing. 
  3. Fig 1, you have to add the scheme of Brain because you described it in the script. Liver-Pancreas-Brain-GUT, since you are mentioning Gut at the end.
  4. Fig 2, should you add IGF-I and II with insulin since you are mentioning, and also add ROS signaling in more details
  5. Fig3: it is a very confusing scheme because of the shape of the arrows. don't use a curved one. what is the meaning of facing the curved arrow and straight-arrow between ROS and Hyperglycemia?
  6. Some of abbreviations are missing, for example NLR, NLRP3 and more
  7. Intense English correction is requiring as you can see from minor comments, there are a lot more that I mentioned below. 

Minor but not least

  1. Line13-14: plasma glucose is increasing after the meal in both normal and diabetic subjects, this is not the reason which triggers prediabetes. explanations?
  2. Line17-18: already described in line15-16, don't need to repeat
  3. Line34: insulin cascade --> insulin signaling cascade
  4. Line43-49: association to Brain pathology is obvious but through which mechanism? and which type of treatment? make the phrase clearer
  5. Line61: not only cardiovascular disease and T2DM.
  6. Line75: delete: as studied in other disease
  7. 82: the detection and management --> diagnosis and treatment
  8. 103: type 1 diabetes --> type 2
  9. 107: feedback is not a proper word in this text
  10. 110: BRAAs
  11. 172: adipose tissue -> adipose tissue mass, produce--> induce
  12. 178: due to
  13. 194: cell metabolic homeostasis
  14. 197: LPS abbreviation
  15. 201: DT2 --> T2DM?
  16. 214: Shc and APS?
  17. 215: ISR-1? --> IRS-1?
  18. 203-220: rephrase, not clear
  19. Akt and AKT are different?
  20. 223-233: rephrase, English correction
  21. 238: TNF-a damages insulin signaling? what damages meaning?
  22. 268-270: Ref?
  23. 284-285: a decrease in the reduced?
  24. 288: Diabetic women and T2DM in the same sentence?
  25. 288-304: women > men: no matter ages
  26. 306-308: men < women ?? it is contradictory and so confusing, rephrase the paragraph of both
  27. 310: what do you mean by saying 'virtually'?? 
  28. 363: what do you mean 'organ perfusion'? 
  29. 368-371: speaking about renal disease is out of focusing
  30. 379-387: rephrase, not clear
  31. 415-417: speaking about atherosclerosis is also out of focusing again
  32. from section 5.1: many phrases are repeating the same story, please rephrase them and make them clear
  33. 460: 'therefore' is not properly used

Round 2

Reviewer 3 Report

The manuscript is improved a lot. 

  1. The authors agreed to modify the title, abstract, and conclusion but I don't see that it changed.
  2. There were still some errors in English writing, ex) because --> because of
